# How Useful Is Preoperative Aspiration before Revision of Unicompartmental Knee Prostheses Because of Osteoarthritis in the Other Compartments?

**DOI:** 10.3390/antibiotics13040361

**Published:** 2024-04-15

**Authors:** Benedikt Paul Blersch, Florian Hubert Sax, Bernd Fink

**Affiliations:** 1Department of Joint Replacement, General and Rheumatic Orthopaedics, Orthopaedic Clinic Markgröningen gGmbH, Kurt-Lindemann-Weg 10, 71706 Markgröningen, Germany; benedikt.blersch@rkh-gesundheit.de (B.P.B.); florian.sax@rkh-gesundheit.de (F.H.S.); 2Orthopaedic Department, University Hospital Hamburg-Eppendorf, Martinistrasse 52, 20246 Hamburg, Germany

**Keywords:** preoperative aspiration, subsequent osteoarthritis, revision of unicompartmental knee arthroplasty

## Abstract

Aim: Periprosthetic joint infections (PJIs) of unicompartmental knee arthroplasties (UKAs) can lead to secondary osteoarthritis of the other compartments. The objective of this study was to identify the frequency of PJIs in cases of UKA with progressed secondary osteoarthritis and the result of septic one-stage revision in these cases to verify the value of preoperative aspiration in cases of secondary osteoarthritis of UKA. Methods: We retrospectively reviewed 97 patients with a unicompartmental arthroplasty who underwent revision surgery to a total knee arthroplasty (TKA) between January 2013 and March 2021 because of subsequent osteoarthritis. Preoperative aspiration and sample collection during the revision surgery were employed to identify potential periprosthetic joint infections (PJIs). The post-revision period was monitored for septic complications over an average duration of 55.7 ± 25.2 months (24–113). Results: PJIs were identified in 5.2% of cases through preoperative aspiration. In all instances of PJIs, a one-stage septic revision was performed, and notably, none of these cases experienced septic complications during the follow-up period. Conclusions: Preoperative aspiration is essential in order to exclude the presence of a PJI before performing revision surgery of UKA due to secondary osteoarthritis.

## 1. Introduction

Unicompartmental knee arthroplasty (UKA) is a successful treatment option for unicompartmental osteoarthritis of the knee. Several reports have demonstrated survival rates greater than 90% at 10 years after modern UKA implantation in clinics with experienced surgeons [1,2,3,4,5]. Advantages in UKA compared to total knee arthroplasty (TKA) such as less invasive surgery, shorter operative time and hospital stay, lower intraoperative blood loss, and higher postoperative range of motion and level of activity, have led to a considerable increase in primary UKA numbers in recent years [6,7,8,9]. Accepted indications for UKA are unicompartmental osteoarthritis of the medial or lateral compartment with no osteoarthritis in the other compartments as well as intact anterior crucial ligament and medial collateral ligament [1,2,5,6,9]. On the other hand, contraindications for unicompartmental knee arthroplasties are flexion contracture of more than 15 degrees, low range of motion with flexion below 100 degrees, septic arthritis of the joint, inflammatory arthritis with activity in the knee, and ligamentous instability, especially rupture of the anterior crucial ligament [1,2,5,6,9]. However, osteoarthritis of another compartment may eventually occur and is one of the most frequent reasons for revision of UKAs [8,10].

Citak et al. identified arthritis within the other compartment as the main cause of UKA revision with a percentage of 39.5 [10]. Pandit et al. researched the frequency of complications after UKA and detected arthritis within the lateral compartment as the most frequent complication in 0.9% [11].

Even periprosthetic joint infection (PJI) is less frequent in UKA than in TKA; it is the reason for revision of UKA in 0.2% to 1% of cases [12,13]. Because chronic periprosthetic joint infection can lead to secondary osteoarthritis of the other compartments [14,15], some revisions due to progression of osteoarthritis may have been the result of unrecognized periprosthetic joint infections. Preoperative diagnostics with aspiration of the joint represent an established method for ruling out or verifying periprosthetic joint infection but are not performed as a routine before revision of UKAs when osteoarthritis has been diagnosed in the other compartments. If a periprosthetic infection would be diagnosed preoperatively, a single-stage septic revision procedure could be performed so that preoperative aspiration as a routine may be helpful to reduce the amount of unrecognized periprosthetic joint infections in failed unicompartmental knee arthroplasties. To the best of our knowledge, there are no publications dealing with this topic of preoperative aspiration for ruling out or detecting periprosthetic joint infection in failed unicompartmental knee arthroplasties because of secondary osteoarthritis of other compartments. Therefore, the aim of the current study was to answer the following questions:
How often is PJI recognized in UKAs where progressed osteoarthritis is given as the reason for revision?What are the success rates of the aseptic and septic one-stage revisions of UKAs to total knee arthroplasties?

## 2. Results

Preoperative aspiration revealed PJIs according to the criteria for a periprosthetic infection based on the ICM score in five (5.2%) patients (Figure 1, Table 1). In all cases, a one-stage septic revision was performed. Intraoperatively collected specimens confirmed PJIs in each case (Figure 1, Table 1). The postoperative course after 90 septic revisions was without recurrence of septic complications in all cases.

Preoperative PJIs were identified in three females and two males with a mean age of 69.4 years; the average body mass index (BMI) was 28.4 kg/m^2^. Regarding the American Society of Anesthesiologists (ASA) score, three patients were categorized as ASA 2 and two patients as ASA 3. The Charlson Comorbidity Index (CCI) classifies one patient each within CCI-Score 1, CCI-Score 2, CCI-Score 4, CCI-Score 5, and CCI-Score 6. In three cases, *Staphylococcus epidermidis,* and in one case, *Cutibacterium acnes* was identified in the preoperative cultivation as well as in the intraoperative culture. Remarkably, in one case, the perioperative cultivation isolated *Staphylococcus epidermidis* and later on, the intraoperative culture showed *Staphylococcus epidermidis* and additionally *Cutibacterium acnes*. Following the ICM score, three patients reached an ICM score of 7 and two patients an ICM score of 10.

In 92 (94.8%) patient cases, the preoperative aspirate was negative for the presence of PJI. Therefore, aseptic revision of the unicompartmental prosthesis was performed (Figure 1).

Because the criteria of a PJI were met in the intraoperative specimens, prolonged postoperative antibiosis was carried out in one patient case (1.0%) (Figure 1, Table 1). In this case, *Staphylococcus epidermidis* was detected within the intraoperative culture. No reinfection occurred in the subsequent follow-up.

One case (1.0%), without evidence of PJI in the preoperative aspirate and intraoperative specimens, experienced a septic complication that was successfully treated with one-stage septic revision (Figure 1).

## 3. Discussion

Routine preoperative aspiration of joints with unicompartmental prostheses that are referred for prosthesis revision due to osteoarthritis revealed a periprosthetic infection in 5.2% of cases. To the best of our knowledge, this is the first study to systematically address this issue.

In previous studies of the literature concerning the outcome of revision surgery of UKAs, Leta et al. [16], analyzed 578 cases from the Norwegian Prosthesis Register and found that periprosthetic infection occurred postoperatively in 16% of cases. For the Australian Prosthesis Registry, Hang et al. [17] found an infection rate after revision surgery from UKAs to total knee arthroplasties (TKAs) of 14%. In a meta-analysis of 1373 revised UKAs, Shen et al. [18] found a rate of PJIs of 4.1%. Due to the lack of preoperative aspiration, it is possible that a PJI exists prior to revision surgery but was not diagnosed preoperatively. Consequently, the PJI could not adequately be addressed with local and systemic antibiotic therapy. Such local and systemic specific antibiotic therapy, if the microorganism had been known preoperatively, would in all likelihood have allowed successful one-stage septic revision and so avoided postoperative periprosthetic infection. This is supported by the favorable outcomes of septic one-stage revisions of UKAs in the studies by Singer et al. [19], with a 100% rate of infection control in 6 cases after a mean follow-up of 36 months (24–72 months) and a rate of infection control of 93.3% after 8 years for 15 patients in a report by Kocaoglu et al. [20]. Like the present study with 100% freedom from infection in five cases after 48.6 ± 29.5 months, they had preoperative knowledge of the pathogen and carried out specific local and systemic antibiotic therapy.

Moreover, if a periprosthetic joint infection as the reason of secondary osteoarthritis in the other components could be ruled out by routine preoperative aspiration, prolonged antibiotic usage can be prevented, which may otherwise be prescribed by some surgeons in unclear situations. By this, the risk of allergic reactions to antibiotics may be reduced also [21].

The study has some limitations. It is a retrospective analysis of data collected prospectively and entered into a database. This may result in a selection bias. The study quality could be further improved by future prospective multicenter studies with a larger number of cases. The number of periprosthetic infections of UKAs is low, which limits the evaluation of the therapeutic success of one-stage septic revisions. However, this is due to the low incidence of such infections in UKAs, and the case number is consistent with other studies [19,20].

Moreover, the accuracy of aspiration alone in diagnosing periprosthetic infection is not 100%. Diagnostic tests with high sensitivities mostly have lower specificities and result in a higher percentage of patients treated for periprosthetic joint infections. The opposite situation exists for tests with lower sensitivities and higher specificities. Fink et al. analyzed the serum C-reactive protein level, the synovial fluid obtained by joint aspiration and five synovial biopsies in 145 cases of knee replacements prior to revision to assess the value of these parameters in diagnosing PJI [22]. They showed a sensitivity of aspiration of 72.5%, a specificity of 95.2%, and an accuracy of 89%. The authors emphasize the diagnostic value of joint biopsy, which had a sensitivity of 100%, a specificity of 98.1%, and an accuracy of 98.6% [22]. Barrack et al. analyzed 78 cases of aspiration in order to detect PJI after TKA and showed a sensitivity of 65.4% and a specificity of 96.1% [23]. By performing several tests simultaneously (C-reactive serum level, leukocyte in serum culturing, leukocyte count in the aspirate, alpha-defensin determination), a very high accuracy can be achieved [24].

Moreover, the findings of the aspiration were confirmed in all cases by the tissue samples taken intraoperatively for culturing and histological study. We also observed one case of PJI which was not detected by the preoperative aspiration. However, in our opinion, this case does not argue against the benefit of preoperative aspiration before revision of a unicompartmental endoprosthesis with osteoarthritis.

## 4. Material and Methods

The study included 97 patients with a unicompartmental arthroplasty (UKA) who underwent revision surgery to a total knee arthroplasty (TKA) between January 2013 and March 2021 because of subsequent osteoarthritis. All patients underwent preoperative aspiration of the associated knee joint. Revisions of unicompartmental prosthesis because of other causes, such as loosening, lack of osteointegration, ligamentous instability, fracture, inlay dislocation, and arthrofibrosis, were excluded from the study because these indications for revision were not the topic of this study. However, preoperative aspirations before revision were also performed in these cases, because preoperative aspiration is performed in our clinic as a routine procedure before any revision surgery.

The cohort consisted of 65 females and 32 males, aged 70.6 ± 9.4 (40.0–98.0) years. There were 87 revisions (89.7%) of a medial unicompartmental prosthesis to bicondylar total knee arthroplasty (TKA), 7 revisions (7.2%) of a lateral unicompartmental prosthesis to TKA, and 3 revisions (3.1%) of a medial unicompartmental prosthesis to hinged TKA because of intraoperative collateral ligament instability. The time between primary surgery and revision surgery was 81.5 ± 47.9 (4.0–197.0) months. With respect to the ASA classification scores, 3 patients were classed as ASA 1, 58 patients were ASA 2, 36 patients ASA 3, and 0 patients ASA 4 [25,26]. Regarding the Charlson Comorbidity Index (CCI), there were 2 patients with CCI 0, 1 patient with CCI 1, 9 patients with CCI 2, 21 patients with CCI 3, 24 patients with CCI 4, 14 patients with CCI 5, 14 patient with CCI 6, 6 patients with CCI 7, 3 patients with CCI 8, 2 patients with CCI 9, and 1 patient with CCI 10 [26,27].

Preoperative aspiration was conducted, and the aspirate was subjected to microbial cultivation, cell count determination, and analysis of alpha-defensin levels (as of 2017). Cell quantification was performed by aspirating a minimum of 1 mL synovial fluid into an EDTA tube, followed by cell count determination using the ABX Pentra XL 80 laboratory diagnostic device (Horiba Medical, Montpellier, France). In addition, the collected fluid was promptly transferred into pediatric blood culture bottles containing BD BACTEC-PEDS-PLUS/F-Medium (Becton Dickinson, Heidelberg, Germany) and underwent a 14-day incubation period [28]. Alpha-defensin levels were assessed via ELISA test. Serum CRP-levels were determined across all cases.

During revision surgery, samples were extracted from five distinct areas close to the prosthesis (periprosthetic tissue and synovium). Additionally, five samples from the synovium and periprosthetic connective tissue membrane associated with the loosened prosthesis were procured for histological evaluation. Perioperative antibiotic administration occurred subsequent to the collection of all samples. Biopsy samples were placed in sterile tubes and promptly transported to the microbiological laboratory, similar to the aspirated fluid, within one hour of sampling. These samples were then streaked onto blood agar and inoculated into special nutrient broth for anaerobic organisms, with an incubation period of 14 days [28]. Tissue analyses and aspiration results were evaluated based on the ICM criteria [29,30,31], whereby a diagnosis of periprosthetic joint infection (PJI) was assigned if the cumulative diagnostic score reached at least 6. Morawietz and Krenn et al.’s classification system [32,33,34] was utilized for histological analysis of periprosthetic tissue to distinguish between wear particle type (I), infection type (II), combined type (III), and indeterminate type (IV). Furthermore, the count of polymorphonuclear leukocytes per high power microscope field was determined.

After the revision surgery, patients were followed up for at least 2 years. The data were collected prospectively in our database and were analyzed retrospectively. The mean follow-up was 55.7 ± 25.2 (24–113) months. In cases of PJI, a septic one-stage revision was performed with specific local antibiotics in the bone cement and systemic antibiotic therapy (2 weeks intravenous and 4 weeks oral) according to the susceptibility of the detected microorganism. Patients were categorized as reinfection-free according to Diaz-Ledezma et al. [35] if they fulfilled the subsequent conditions: absence of PJI-related mortality, absence of further PJI-related surgeries, and both microbiological and clinical absence of infection for a minimum duration of 24 months. An internal CRP detection threshold of ≥10 mg/L was established [35].

Statistical analysis utilized SPSS for Windows (version 22; IBM Corp.; Armonk, NY, USA). All participants provided informed consent prior to participation. Unless otherwise stated, descriptive results are expressed as mean ± standard deviation (and range) or absolute number (percentage), respectively. The level of significance was set at *p* < 0.05. The study was conducted in accordance with the Declaration of Helsinki, and the protocol was approved by the Ethics Committee of Landesärztekammer Badenwürttemberg (committee’s reference number F-2023-115).

## 5. Conclusions

In summary, we suggest that the incidence of periprosthetic infection of 5.2% associated with osteoarthritis subsequent to UKA and the successful treatment of the PJI with a single-stage septic revision represents a clear indication of the benefit to be gained by routinely aspirating the joint requiring revision of a UKA due to subsequent osteoarthritis. We therefore recommend that before every revision of a UKA, the affected joint should be aspirated in order to exclude the presence of a periprosthetic infection.

## Figures and Tables

**Figure 1 antibiotics-13-00361-f001:**
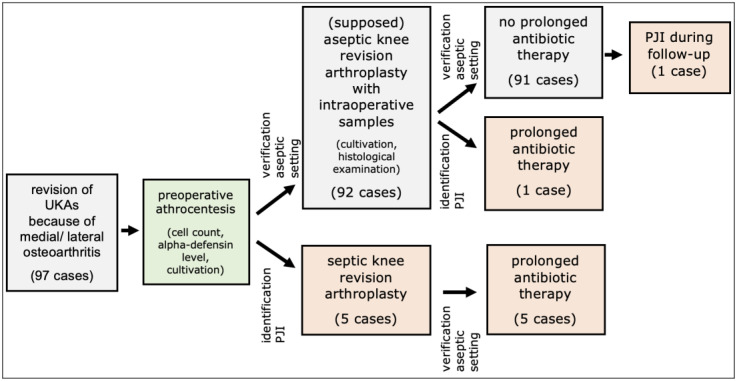
Patient flowchart. PJI = periprosthetic joint infection.

**Table 1 antibiotics-13-00361-t001:** Parameters of the 5 patients with preoperative detected periprosthetic joint infection (PJI) (1 to 5) and 1 patient (6) with preoperative unrecognized PJI. mo = months, HPF = high power field (×400).

	Gender	Age	BMI	ASA	CCI	SERUM Preop. CRP (mg/L)	SERUM Preop. Leukocyte (n/µL)	SYNOVIA Preop. Cell Count (n/µL)	SYNOVIA Preop. Alpha Defensin Level (ng/mL)	SYNOVIA Preop. Cultivation	Preop. ICM-Score (Including Preop. Cultivation)	Antibiosis i.v. (2 Weeks)	Antibiosis p.o. (4 Weeks)	Antibiosis Intraop. Cement	Intraop. Culture	Histology Type Morowitz/ Krenn	Histology Neutrophils per HPF	Follow-Up (mo)
1	female	80	32.4	2	5	10.1	5.3	1500	1.1	*Cutibacterium acnes*	7	Penicillin G i.v., Rifampicin p.o.	Levofloxacin p.o., Rifampicin p.o.	Copal G+C	*Cutibacterium acnes* (3/5)	II	20	84
2	male	62	27.8	2	2	10.3	9.7	2540	1.5	*Staphylococcus epidermidis*	7	Flucloxacillin i.v.	Amoxicillin p.o.	Copal G+C	*Staphylococcus epidermidis* (5/5), *Cutibacterium acnes* (2/5)	III	5	30
3	male	71	26.4	3	6	11.7	69.8 (CLL)	26,000	1.6	*Staphylococcus epidermidis*	10	Vancomycin i.v., Rifampicin p.o.	Linezolid p.o.	Copal G+V	*Staphylococcus epidermidis* (2/5)	III	>50	89
4	female	76	24.4	3	4	18.1	7.1	27,200	2.1	*Staphylococcus epidermidis*	10	Cefuroxim i.v., Rifampicin p.o.	Levofloxacin p.o., Rifampicin p.o.	Copal G+C	*Staphylococcus epidermidis* (2/5)	II	>100	26
5	female	58	31.2	2	1	25.0	7.94	2350	1.6	*Staphylococcus epidermidis*	7	Cefuroxim i.v., Rifampicin p.o.	Levofloxacin p.o., Rifampicin p.o.	Copal G+C	*Staphylococcus epidermidis* (5/5)	II	>30	27
6	male	78	29.4	2	3	22.1	11.0	400	<0.1	no cultural growth	2	Cefuroxim i.v., Rifampicin p.o.	Levofloxacin p.o., Rifampicin p.o.	Copal G+C	*Staphylococcus epidermidis* (2/5)	III	13	36

## Data Availability

We do not wish to share our data because of some of patients’ data regarding individual privacy, and according to the policy of our hospital, the data could not be shared to others without permission.

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
