# Peer review of "How Useful Is Preoperative Aspiration before Revision of Unicompartmental Knee Prostheses Because of Osteoarthritis in the Other Compartments?"

_antibiotics, 2024, doi:10.3390/antibiotics13040361_

Round 1

Reviewer 1 Report

Comments and Suggestions for Authors

The manuscript entitled "How useful is preoperative aspiration before revision of unicompartimental knee prostheses because of osteoarthritis in the other compartments?" is a retrospective study reporting the results of 97 patients with a unicompartmental arthroplasty who underwent revision surgery to a total knee arthroplasty to answer two questions:

1. How often is PJI recognized in UKAs where progressed osteoarthritis is given as the reason for revision? 

2. What are the success-rates of the aseptic and septic one-stage revisions of UKAs to total knee arthroplasties? 

The number of patients included in this study is low, and such limittions were discussed in the manuscript.  

However, the relevance of the study with the scope of the journal Antibiotics is not clear. Also, the novelty of the study was not mentioned.

Moreover, the Latin names of the bacteria should be written italic. 

Author Response

The number of patients included in this study is low, and such limitations were discussed in the manuscript.  No correction necessary

However, the relevance of the study with the scope of the journal Antibiotics is not clear. Also, the novelty of the study was not mentioned. This is mentioned in more detai at the end of the introduction section: If a periprosthetic infection would be diagnosed preoperatively a single-stage septic revision procedure could be performed, so that preoperative aspiration as a routine may be helpful to reduce the amount of unrecognized periprosthetic joint infections in failed unicompartmental knee arthroplasties. To the best of our knowledge, there are no publications dealing with this topic of preoperative aspiration for ruling out or detecting periprosthetic joint infection in failed unicompartmental knee arthroplasties because of secondary osteoarthritis of other compartments.

Moreover, the Latin names of the bacteria should be written italic. This is corrected

Reviewer 2 Report

Comments and Suggestions for Authors

the article is interesting and the topic covered is very current. The language is adequate. 

The correct order of sections is introduction, materials and methods, results, discussion, conclusion. Authors need to re-order the sections in the text.

Statistical analysis had to be better explained. 

Minor revision

Author Response

The article is interesting and the topic covered is very current. The language is adequate. 

The correct order of sections is introduction, materials and methods, results, discussion, conclusion. Authors need to re-order the sections in the text. The order of the sections are according to the style of Antibiotics. If the editor or the journal wish we will of course change the order.

Statistical analysis had to be better explained. This is done at the end of the Material and Method section: Statistical analysis utilized SPSS for Windows (version 22; IBM Corp.; Armonk, NY). All participants provided informed consent prior to participation. Unless otherwise stated, descriptive results are expressed as mean ± standard deviation (and range) or absolute number (percentage), respectively. The level of significance was set at p < 0.05.

Reviewer 3 Report

Comments and Suggestions for Authors

1. the introduction section should also include selection criteria and situations where UKA might not be the best option

2. the introduction should seek to explain the limitations of current diagnostic methods for PJI in UKA

3. discuss some of the patient selection criteria, surgeon experience, and implant selection that make UKA successful

4. each exclusion criteria should be explained

5. provide a statistical analysis that accounts for varying follow-up times

6. what are some preoperative aspiration any thresholds or indicators used as an indication

7. Is there any selection bias?

8. discussion should encompass how does sensitivity, specificity, and accuracy of different diagnostic tests affect clinical decision

9. how can the retrospective design might influence the results and the potential biases?

References are new, results are ok

Author Response

  1. the introduction section should also include selection criteria and situations where UKA might not be the best option. This is added in the introduction section: Contraindications for unicompartmental knee arthroplasties are flexion contracture of more than 15 degrees, low range of motion with flexion below 100 degrees, septic arthritis of the joint, inflammatory arthritis with activity in the knee and ligamentous instability, especially rupture of the anterior crucial ligament [1,2,5,6,9].

  1. the introduction should seek to explain the limitations of current diagnostic methods for PJI in UKA. The limitations of the diagnostic tests are mentioned in the limitation section. Because the aim of the study was not to analyze the value of diagnostic tests the limitations are mentioned in the limitation section. If the editor and reviewer wish we can also add that in the introduction section.

  1. discuss some of the patient selection criteria, surgeon experience, and implant selection that make UKA successful. This is explained in more detail in the introduction section: Several reports have demonstrated survival rates greater than 90% at 10 years after modern UKA implantation in clinics with experienced surgeons [1-5]. Accepted indications for UKA are unicompartmental osteoarthritis of the medial or lateral compartment with no osteoarthritis in the other compartments as well as intact anterior crucial ligament and medial collateral ligament [1,2,5,6,9].

  1. each exclusion criteria should be explained. The explanation for the exclusions is added in the Material and Method section: Revisions of unicompartmental prosthesis because of other causes, such as loosening, lack of osteointegration, ligamentous instability, fracture, inlay dislocation, and arthrofibrosis, were excluded from the study because these indications for revision were not the topic of this study. However, preoperative aspiration before revision were also performed in these cases, because preoperative aspiration is performed in our clinic as a routine procedure before any revision surgery.

  1. provide a statistical analysis that accounts for varying follow-up times. The statistical analyses are described in more detail at the end of the Material and Method section: Statistical analysis utilized SPSS for Windows (version 22; IBM Corp.; Armonk, NY). All participants provided informed consent prior to participation. Unless otherwise stated, descriptive results are expressed as mean ± standard deviation (and range) or absolute number (percentage), respectively. The level of significance was set at p < 0.05.

  1. what are some preoperative aspiration any thresholds or indicators used as an indication. The indication of preoperative aspiration is explained in more detail in the Material and Method section. It is a routine procedure before any revision surgery is performed in total joint arthroplasties in our clinic: “However, preoperative aspiration before revision were also performed in these cases, because preoperative aspiration is performed in our clinic as a routine procedure before any revision surgery.”

  1. Is there any selection bias? This is added in the limitations: The study has some limitations. It is a retrospective analysis of data collected prospectively and entered into a database. This may result in a selection bias.

  1. discussion should encompass how does sensitivity, specificity, and accuracy of different diagnostic tests affect clinical decision. Is discussed in the limitations: Moreover, the accuracy of aspiration alone in diagnosing periprosthetic infection is not 100%. Diagnostic tests with high sensitivities mostly have lower specificities and result in a higher percentage of patients treated for periprosthetic joint infections. The opposite situation exists for tests with lower sensitivities and higher specificities.

  1. how can the retrospective design might influence the results and the potential biases? This is added in the limitations: The study has some limitations. It is a retrospective analysis of data collected prospectively and entered into a database. This may result in a selection bias.

References are new, results are ok No comments necessary

Reviewer 4 Report

Comments and Suggestions for Authors

Dear Authors,

I read with great interest the article about the presence of infections associated to knee prosthesis replacement.

However, there are some aspects that require your attention.

In Table 1, you need to remove the line between lines 5 and 6.

In Figure 1, you need to explain the PFJ abbreviation.

In the discussions section you need to underline also that give the negative presence of infection the case management could benefit from a reduction of the use of antibiotics. This also could diminish the risk of allergic reactions to antibiotics. Please reference this to the article by  Ovidiu B, Dumitru M, Vrinceanu D, Cergan R, Jeican II, Giurcaneanu C, Miron A: Current approach to medico-legal aspects of allergic reactions. Rom J Leg Med. 2021, 29:328-31. 10.4323/rjlm.2021.328

Moreover, you have many abbreviations in the text, you need to insert a list of abbreviations at the end of the manuscript.

Looking forward to receiving the improved version of your manuscript.

Author Response

However, there are some aspects that require your attention.

In Table 1, you need to remove the line between lines 5 and 6. Is done

In Figure 1, you need to explain the PFJ abbreviation. Is corrected to PJI

In the discussions section you need to underline also that give the negative presence of infection the case management could benefit from a reduction of the use of antibiotics. This also could diminish the risk of allergic reactions to antibiotics. Please reference this to the article by  Ovidiu B, Dumitru M, Vrinceanu D, Cergan R, Jeican II, Giurcaneanu C, Miron A: Current approach to medico-legal aspects of allergic reactions. Rom J Leg Med. 2021, 29:328-31. 10.4323/rjlm.2021.328 Is done: Moreover, if a periprosthetic joint infection as the reason of secondary osteoarthritis in the other components could be ruled out by routine preoperative aspiration, prolonged antibiotic usage can be prevented, which may otherwise be prescribed by some surgeons in unclear situations. By this the risk of allergic reactions to antibiotics may be reduced also [19].   

Moreover, you have many abbreviations in the text, you need to insert a list of abbreviations at the end of the manuscript. Is done